# Can Hepatitis B Virus (HBV) Reactivation Result from a Mild COVID-19 Infection?

**Ioannis Braimakis** [1], **Sofia Vasileiadi** [2], **Eleni-Myrto Trifylli** [3], **Nikolaos Papadopoulos** [4,*] **and Melanie Deutsch** [2]

1   Gastroenterology Department, 417 Army Share Fund Hospital of Athens, 11521 Athens, Greece; giannis_br@hotmail.com
2   2nd Department of Internal Medicine, Hippokration General Hospital of Athens, Medical School of National & Kapodistrian University of Athens, 11527 Athens, Greece; vasileiadi.sofia@gmail.com (S.V.); meladeut@gmail.com (M.D.)
3   1st Department of Internal Medicine, 417 Army Share Fund Hospital of Athens, 11521 Athens, Greece; trif.lena@gmail.com
4   2nd Department of Internal Medicine, 401 General Army Hospital of Athens, 11525 Athens, Greece
*   Correspondence: n.m.papadopoulos@army.gr; Tel.: +30-6932604019

**Abstract:** Hepatitis B virus reactivation (HBVr) is a well-described result of immunosuppressive therapy initiation in various diseases, with the dose and duration of treatment being the main factors determining the probability for reactivation. Such cases have also been described in COVID-19 patients treated with immunosuppressive therapies. Nevertheless, cases of COVID-19 infection that led to HBVr with no concurrent immunosuppressive treatment or any other related cause have also been reported. By that observation, we present a patient followed for a period spanning 20 years with HBeAg negative chronic HBV infection and non-detectable HBV DNA who, after a mild COVID-19 infection treated only with low-dose and short-duration-inhaled corticosteroids (ICS), developed elevated AST and ALT as well as elevated HBV DNA levels. Other etiologies of abnormal liver biochemistries during the diagnostic workout were excluded; thus, the diagnosis of HBV reactivation was established. Treatment with entecavir was initiated, leading to the normalization of AST and ALT levels and a decreasing trend of HBV DNA levels. Since other causes of reactivation were excluded, and the ICS dose and duration were found baring only a very low risk (<1%) for HBVr, COVID-19 infection could be considered the most probable cause of reactivation, hence underlining the need for the close monitoring of those patients.

**Keywords:** HBV reactivation; COVID-19; immunosuppressive therapy

## 1. Introduction

Hepatitis B virus (HBV), part of the Hepadnaviridae family [1], is responsible for about 296 million chronic hepatitis infections according to the latest World Health Organization (WHO) data [2]. Following HBV infection and based on the interaction between HBV and the host's immune system response, the acute infection can either resolve or progress to chronic hepatitis B [1]. Current nomenclature for HBV chronic infection marks the presence of five separate phases, not always as an infection sequence, but based on HBeAg seropositivity as well as serum levels of HBV DNA and alanine aminotransferase (ALT) [1]. Thus, the following phases arise: HBeAg (+) chronic HBV infection, HBeAg (+) chronic hepatitis B, HBeAg (−) chronic HBV infection, HBeAg (−) chronic hepatitis B and the HBsAg-negative phase [1,3]. Common ground in chronic Hepatitis B or infection, regardless of the phase, is the persistent presence of the covalently closed circular DNA (cccDNA) inside the nucleus of the hepatocyte [4]. The cccDNA acts as the template for HBV transcripts that are subsequently translated into the main proteins of the virus [1,5] and plays a vital role in HBV reactivation (HBVr) at any time and any phase, especially under conditions of immunosuppression [6–9].

As of late 2019, the coronavirus disease 2019 (COVID-19) pandemic has proved to be a serious global public health issue. The responsible virus itself (severe acute respiratory syndrome coronavirus (2-SARS-CoV-2)) not only affects the respiratory system, causing upper and lower respiratory tract infections, but, as already reported in several studies, it also causes liver injury with abnormal liver biochemistry [10–13]. Furthermore, liver dysfunction, as indicated by abnormal liver biochemistries, correlates with COVID-19 infection severity [12]. Thus, increased levels of AST, ALT and bilirubin and decreased albumin levels are noted in patients with severe clinical manifestations of COVID-19 [12]. Furthermore, liver injury in COVID-19 patients has also been associated with worse disease outcomes [11]. Multiple possible liver injury mechanisms have been proposed in the literature. Such a pathophysiological mechanism is the direct cytopathic effect of SARS-CoV-2 through binding to the angiotensin-converting enzyme 2 (*ACE2*) and transmembrane serine protease 2 (*TMPRSS2*) receptors of cholangiocytes and hepatocytes [14]. Other possible mechanisms described in the literature include liver injury caused by severe inflammation; hypoxia, especially in cases of severe respiratory tract infection; the toxicity of the drugs used for COVID-19 treatment; and vascular changes attributed to coagulopathy [15,16].

Taking into consideration the fact that both HBV and SARS-CoV-2 can potentially cause liver injury, several studies attempted to investigate coinfection cases and highlight the effect of the latter on chronic HBV natural history on one hand, and the effect of HBV seropositivity on the severity of COVID-19 infection on the other hand [17]. Attempting to help address this intriguing matter, we report a case of a patient presenting with HBV reactivation soon after COVID-19 infection without the administration of systemic immunosuppressive medication.

## 2. Case Report

A 49-year-old immunocompetent female presented in an outpatient hepatology department with elevated serum aspartate aminotransferase (AST) and ALT levels. The patient has been followed for HBeAg-negative chronic HBV infection every 6 months for 20 years, presenting normal liver enzymes and undetectable HBV DNA. The patient's medical history included neither health problems needing long-term medical treatment nor diseases affecting her immune system, such as diabetes mellitus. Moreover, no alcohol consumption, medications over the counter or consumption of herbs or smoking were mentioned. She also reported no past surgical history.

The patient reported symptoms indicative of respiratory tract infection (fatigue, dry cough, sore throat and anosmia) 3 weeks before her presentation at the hepatology department. Since she mentioned close contact with a COVID-19-positive patient, she was tested for COVID-19 infection with a polymerase chain reaction (PCR) test on the second day of her symptoms. The PCR test for COVID-19 came out positive, and due to mild respiratory manifestations without fever or hypoxia, only a combination of budesonide, 160 μg, plus formoterol fumarate dehydrate, 4.5 μg, was prescribed. The treatment duration was 5 days, with a resolution of her symptoms by day 4 of treatment initiation. Over the course of the SARS-CoV-2 infection, the patient experienced no other symptoms or complications and did not need hospitalization. Thus, there was no need for further COVID-19 treatment with systemic corticosteroids such as dexamethasone, baricitinib (Janus Kinase inhibitor) or tocilizumab (anti-interleukin-6 receptor (IL-6R) antagonist) [18].

At her first hepatology consultation appointment 3 weeks after the COVID-19 infection, the patient experienced no other symptoms besides abnormal liver chemistries. More specifically, in the blood tests presented by the patient, aminotransferases were mildly elevated, with an AST level double the upper limit of normal (ULN) and ALT level triple the ULN (Table 1). No other vital deviations from normal were noted in the blood tests presented (Table 1). Liver and hepatitis panel tests and upper abdominal ultrasound were prescribed, and at her second consultation appointment, the patient presented with the results 1 week later. Persisting elevated AST and ALT levels, double and triple the ULN,

respectively, were observed (Table 1). As for her Hepatitis B status, the patient was HBsAg (+), HBeAg (-), anti-HBe (+) and anti-HBc (+), with HBV-DNA at 9,350,000 iu/mL (Table 2).

**Table 1.** The trend over time of liver function tests (LFTs).

| | First Consultation (3 Weeks after COVID-19 Infection) | Second Consultation (4 Weeks after COVID-19 Infection) | Two Months after Entecavir, 0.5 mg |
|---|---|---|---|
| AST (U/L) | 82 | 71 | 28 |
| ALT (U/L) | 106 | 90 | 30 |
| ALP (U/L) | 106 | 80 | 84 |
| γGT (U/L) | 16 | 19 | 20 |

**Table 2.** The trend over time of HBV serologic markers, anti-HCV and anti-HDV.

| | Prior 6 Months from COVID-19 Infection | Second Consultation (4 Weeks after COVID-19 Infection) | Two Months after Entecavir, 0.5 mg |
|---|---|---|---|
| HBsAg | + | + | |
| HBeAg | − | − | |
| Anti-HBc | + | + | |
| Anti-HBe | + | + | |
| Anti-HCV | − | − | |
| Anti-HDV | − | − | |
| HBV DNA (IU/mL) | Non-Detectable | 9,350,000 | 54,000 |
| IgG (mg/dL) | | 1555 | |

Moreover, the patient tested negative for anti-HCV antibodies, anti-HDV antibodies, antinuclear antibodies (ANA), antimitochondrial antibodies (AMA), anti-smooth-muscle antibodies (ASMA) and anti-dsDNA antibodies (Table 2). Immunoglobulin levels showed no deviation from the average (Table 2). The upper abdominal ultrasound revealed no abnormalities from her liver, gallbladder, biliary tree, pancreas or spleen, while alpha-fetoprotein (a-FP) levels came out normal. Consequently, abnormal liver chemistries were attributed to HBV reactivation, and thus entecavir at a dose of 0.5 mg once daily was prescribed.

Two months after initiating treatment with entecavir, the patient presented for a regular hepatology consultation. Laboratory evaluation revealed that AST and ALT levels had normalized, and no other deviation from normal was noted in the LFTs (Table 1). As for HBV DNA levels, they demonstrated a decreasing trend (HBV DNA = 54,000 IU/mL) (Table 2). The patient remains without clinical symptoms and is under close surveillance in the hepatology department.

## 3. Discussion

We present a case of HBV reactivation (HBVr) in an immunocompetent female patient with a 20-year history of chronic HBeAg-negative HBV infection occurring after a mild COVID-19 infection. Usually, HBVr occurs when the host immune system loses the ability to suppress virus replication [19]. Both HBsAg-positive/anti-HBc-positive and HBsAg-negative/anti-HBc-positive patients preserve the potential of HBVr [20] under certain circumstances. These comprise the concomitant use of either immunosuppressive therapies or variable viral interactions. Such interaction is well described in HBV/HCV coinfection cases, where HCV usually suppresses HBV replication, leading to low levels of HBV DNA [21–24]. Consequently, a Direct-Acting antiviral (DAA) therapy aiming for a sustained virologic response (SVR) of HCV infection may disrupt the balance between HBV and HCV, leading to HBV reactivation in cases where no concurrent nucleoside/nucleotide analogue (NA) HBV therapy is applied [21,22].

Regarding the definition of HBVr, various variants have been proposed and used in different studies [25]. We used the latest proposed nomenclature presented by Papatheodoridis et al. (2022) in their latest systematic review, meta-analysis and expert opinion [6]. More specifically, for HBsAg-positive patients with previously undetectable HBV DNA, an increase in HBV DNA $\geq$ 1000 IU/mL is consistent with HBVr [6]. Accordingly, our patient with a known chronic HBV infection (HBsAg-positive) diagnosed 20 years ago with undetectable HBV DNA until her COVID-19 infection that was currently found with elevated levels of HBV DNA met the proposed criteria for HBVr (Table 2) [6].

In our patient, to ensure the accuracy of the HBVr diagnosis, other causes of viral hepatitis such as HAV, HEV, HCV and HDV were excluded. Moreover, a diagnostic workup for other etiologies of abnormal liver biochemistry, such as autoimmune hepatitis (AIH) and Primary Biliary Cholangitis (PBC), using non-organ-specific autoantibodies such as ANA, ASMA and AMA was carried out [26–28]. Imaging with upper abdominal ultrasound was also used to rule out focal liver lesions as a cause for elevated AST and ALT in combination with normal levels of a-FP [29,30].

Following establishing the diagnosis of HBVr, the main question concerned the pathophysiology of HBVr 20 years after the initial diagnosis. The main reason leading to impaired ability to suppress HBV replication and thus to a possible reactivation is the initiation of immunosuppressive treatment [6,25]. A broad spectrum of immunosuppressants have been incriminated for HBVr, such as immune checkpoint inhibitors, tyrosine kinase inhibitors, cytokine inhibitors, CAR T-cell immunotherapy, alkylating agents, anti-proliferative agents, calcineurin inhibitors (CNI), mTOR inhibitors and Janus Kinase inhibitors [6,25]. Additionally, corticosteroid treatment for any indication, including COVID-19, has been in the spotlight of studies investigating its ability to induce HBVr [31–33]. This ability is brought about by the activation of a DNA sequence known as Glucocorticoid Response Element (GRE) enclosed in the HBV genome on the one hand [34] and by immunosuppressive properties of corticosteroids on the other hand [25]. Nevertheless, reactivation probability depends on corticosteroids' dose and treatment duration [6,25]. The American Gastroenterological Association (AGA) addressed this matter [25] by categorizing the HBVr risk induced by immunosuppressants as high when the risk of reactivation is expected to be more than 10%, moderate when the frequency of reactivation is between 1% and 10% and low when reactivation risk is expected to be less than 1%. Thus, for HBsAg-positive patients, a moderate (10–20 mg) or high (>20 mg) dose of prednisone (or equivalent) for $\geq$4 weeks results in an increased risk for HBVr and a low dose (<10 mg) of prednisone (or equivalent) for $\geq$4 weeks results in a moderate risk for HBVr, while corticosteroid therapy for $\leq$1 week results in a low risk for HBVr [25]. Moreover, a dose >20 mg for a treatment duration of >2 weeks results in significant immunosuppression [35].

In our patient, as already mentioned, the course of COVID-19 infection was mild, and as a result, only inhaled budesonide, 160 µg, plus formoterol fumarate dehydrate, 4.5 µg, for a 5-day duration was prescribed. Could this possibly be the reason for HBVr in our case? To address this hypothesis, we had to investigate the possible immunosuppressive properties of inhaled corticosteroids (ICSs). Data regarding ICSs derive from studies addressing their efficacy and safety profile, especially in asthma and chronic obstructive pulmonary disease. ICSs' action is mainly applied locally at the airway level and is effective in low doses [36]. Apart from this, according to a study from Maijers et al. (2020), an increase of 1000 µg of fluticasone propionate is commensurate with 5 mg of prednisone, while 1000 µg of budesonide is commensurate with 2 mg of prednisone [37]. Thus, even the high doses used in severe respiratory disorders, such as a budesonide dose of up to 1600 µg/day or a fluticasone propionate dose of up to 1000 µg/day [37], correspond to a low dose (<10 mg) of prednisone. As a result, the HBVr risk with ICS courses administered for less than 1 week, as in our patient, is lower than 1%.

Since the immunosuppressive properties of the ICS administered fell short of explaining our patient's sudden HBVr, a literature review was conducted to investigate the possibility of HBVr induced by COVID-19. Most cases of HBVr after SARS-CoV-2 infection

were attributed to immunosuppressive treatment such as dexamethasone at 6 mg per day [4], other systemic corticosteroids like methylprednisolone at 40 mg per day [17,38] or tocilizumab (anti-IL-6R antagonist) that was used in severe cases of COVID-19 [18]. The aforementioned systemic corticosteroid doses correspond to a high dose of prednisone (40 mg of prednisone is equivalent to dexamethasone and >40 mg of prednisone is equivalent to methylprednisolone) [39] that, even when administered for <7 days, can result in increasing the risk for hepatitis flare [32,40].

Interestingly enough, we also came across patient cases where COVID-19 was considered to be the trigger for HBVr, with other causes of reactivation being excluded [15,17,41,42]. The pathogenesis pathway for this reactivation in patients with HBV and SARS-CoV-2 coinfection remains to be elucidated. A possible explanation could be that of viral interactions between the Hepatitis B virus and SARS-CoV-2 resembling the interactions already described in cases of HBV/HCV coinfection [23,24]. Furthermore, another proposed mechanism is disrupting the balance of immune system activity on the one hand and HBV replication on the other [15].

## 4. Conclusions

In conclusion, due to the low risk for HBVr (lower than 1%) provoked by ICS and other cases presented, our patient's HBVr was attributed to the recent COVID-19 infection. This deduction underlines the need for close monitoring of patients who are HBsAg-positive in case of infection with SARS-CoV-2 using LFTs and HBV DNA levels. The latter is necessary, especially in elevated LFTs such as AST and ALT. As highlighted in our case presentation, this proposed practice should be applied even when no systemic immunosuppressive treatment is administered. As for the pathophysiological pathway of the reactivation and whether HBVr may result from viral interaction with a non-hepatotropic virus like SARS-CoV-2, more research is needed.

**Author Contributions:** Authors contributed equally to this manuscript. Conceptualization, I.B., N.P. and M.D.; validation, I.B., N.P. and M.D.; investigation, I.B., S.V. and E.-M.T.; resources, I.B.; data curation, I.B.; writing—original draft preparation, I.B., S.V. and E.-M.T.; writing—review and editing, N.P. and M.D.; visualization, I.B.; supervision, M.D. All authors have read and agreed to the published version of the manuscript.

**Funding:** This research received no external funding.

**Institutional Review Board Statement:** Not applicable.

**Informed Consent Statement:** Written informed consent was obtained from the patient to publish this paper.

**Data Availability Statement:** Data are contained within the article.

**Conflicts of Interest:** The authors declare no conflict of interest.

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
