# Peer review of "Can Hepatitis B Virus (HBV) Reactivation Result from a Mild COVID-19 Infection?"

_livers, doi:10.3390/livers3030026_

Round 1

Reviewer 1 Report

Only minor typing errors should be corrected.

Author Response

We thank reviewer 1 for his comments.

  • To address to Reviewer’s concerns about minor typing errors in need of correction (Reviewer 1) we had our manuscript checked by 2 colleagues who are fluent in English writing.

Reviewer 2 Report

Thank you for the interesting case report. The plausibility is sound - and COVID 19 may well have triggered the flare.

1. The one issue is whether the patient may have taken any other medication perhaps given her duration of symptoms (3 weeks) of not feeling well due to the SARS Cov-2 infection. The duration of symptoms of 3 weeks is also unusual given what we know about the incubation period of SARS Cov 2 - however the relative immune suppression of COVID does longer beyond the initial 2 weeks acute infection period. COVID itself produces liver enzyme changes - but rightfully the HBV viral load increased significantly that supports HBV as the cause and not COVID. The mechanism of HBV reactivation after SARS-CoV-2 infection is mainly due to a disruption of the balance between the immune status of the host and viral replication. The question I ask is whether any other factor immune compromised the patient perhaps? No mention is made if patient was diabetic? This should please be noted

2. De novo flares of HBeAg negative HBV are driven by changes in the pre-core and BCP regions of the HBV genome - these would likely have been present for some time. This would have been an interesting question and data in this patient, if available.

The stylistic use of English is not optimal and warrants review

Author Response

We thank the reviewer's 2 comments that have improved our manuscript.

To address to need for moderate editing of English language (Reviewer 2 ) we had our manuscript checked by 2 colleagues who are fluent in English writing. 

We also read Reviewer’s 2 comments with great interest. According to the second reviewer’s first comment, there was a misunderstanding concerning the duration of the Covid-19 infection that we tried to clear up by stating the exact day of symptoms onset, Covid-19 testing, treatment duration and symptoms’ resolution (lines 77-84). Additionally, with respect to the following comment of reviewer 2 we noted that our patient was immunocompetent (line 69) and that no other factor such as diabetes melitus immunocompromised her (line 74). Unfortunately, no data about changes in the pre-core and BCP regions of the HBV genome were available in order to address this matter.

Reviewer 3 Report

The authors have well presented the case report and have discussed the causes and concerns relating to Hepatitis B relapse due to Covid 19 infection nicely. My only concern is the writing style. The language requires editing.

The title of the manuscript needs to be improved too.

The title can be improved.

.....with no need of immunosuppressive therapy can be removed from the title or can be reframed.

Line 56: The word "controversial" does not fit with  the context. 

etc.

Author Response

We thank reviewer 3 for his comments that improved our work.

To address to need for moderate editing of English language (Reviewer 3 ) we had our manuscript checked by 2 colleagues who are fluent in English writing. 

As for Reviewer’s 3 comments we decided to change the title of our article to “Can Hepatitis B virus (HBV) reactivation be the result of a mild Covid-19 infection?” and we also changed the word “controversial” to “intriguing” (line 65).

Reviewer 4 Report

The authors of the paper described and commented on a very interesting case of HBV reactivation as a consequence of COVID-19 infection.

I have not meet such a case personally or in the literature, despite my extensive experience with both COVID and HBV patients.

I suggest that the authors indicate in the case report that the patient did not use tocilizumab or barycytynib (she was not hospitalized, so I guess it was not used), which may be the cause of HBV reactivation.

In addition, please clearly indicate when after COVID-19 the reactivation took place and when the next hepatological consultations took place.

Author Response

We thank reviewer 4 for his comments.

In an effort to reply to the concerns of Reviewer 4 we underlined that no baricitinib or tocilizumab (lines 85-86) was applied to our patient. Moreover, we tried to clarify the timing when the HBVr took place (line 88) and the hepatology consultations took place (line 88, line 94 and table 1)